# The Synthesis of Sponge-like V_2_O_5_/CNT Hybrid Nanostructures Using Vertically Aligned CNTs as Templates

**DOI:** 10.3390/nano14020211

**Published:** 2024-01-18

**Authors:** Matías Picuntureo, José Antonio García-Merino, Roberto Villarroel, Samuel A. Hevia

**Affiliations:** 1Instituto de Física, Pontificia Universidad Católica de Chile, Av. Vicuña Mackenna 4860, Santiago 6904411, Chile; mipicuntureo@uc.cl; 2Centro de Investigación en Nanotecnología y Materiales Avanzados, CIEN-UC, Pontificia Universidad Católica de Chile, Av. Vicuña Mackenna 4860, Santiago 6904411, Chile; 3Departamento de Mecánica, Facultad de Ingeniería, Universidad Tecnológica Metropolitana, Av. José Pedro Alessandri 1242, Ñuñoa 7800003, Chile; jgarciam@utem.cl; 4Departamento de Física, Facultad de Ciencias Naturales, Matemática y del Medio Ambiente, Universidad Tecnológica Metropolitana, Las Palmeras 3360, Ñuñoa 7800003, Chile; 5Millennium Institute on Green Ammonia as Energy Vector—MIGA, Pontificia Universidad Católica de Chile, Av. Vicuña Mackenna 4860, Santiago 6904411, Chile

**Keywords:** vertically aligned carbon nanotubes, vanadium pentoxide, chemical vapor deposition, electron beam deposition, thermal oxidation

## Abstract

The fabrication of sponge-like vanadium pentoxide (V_2_O_5_) nanostructures using vertically aligned carbon nanotubes (VACNTs) as a template is presented. The VACNTs were grown on silicon substrates by chemical vapor deposition using the Fe/Al bilayer catalyst approach. The V_2_O_5_ nanostructures were obtained from the thermal oxidation of metallic vanadium deposited on the VACNTs. Different oxidation temperatures and vanadium thicknesses were used to study the influence of these parameters on the stability of the carbon template and the formation of the V_2_O_5_ nanostructures. The morphology of the samples was analyzed by scanning electron microscopy, and the structural characterization was performed by Raman, energy-dispersive X-ray, and X-ray photoelectron spectroscopies. Due to the catalytic properties of V_2_O_5_ in the decomposition of carbonaceous materials, it was possible to obtain supported sponge-like structures based on V_2_O_5_/CNT composites, in which the CNTs exhibit an increase in their graphitization. The VACNTs can be removed or preserved by modulating the thermal oxidation process and the vanadium thickness.

## 1. Introduction

Vanadium pentoxide is a widely studied material due to its potential impact in several applications, such as gas sensing [1], electrochromic devices [2], photocatalysis [3], solar cells [4], and also in energy storage such as Li/Na/K/Al ion batteries or supercapacitors [5,6,7,8,9]. V_2_O_5_ exhibits the highest oxidation state of vanadium and the most stable one. It has an orthorhombic phase with space group Pmmn, composed of a distorted trigonal bipyramidal coordination polyhedral of O atoms surrounding a central V atom. This configuration gives a layered-like structure, which has attracted interest due to its high surface area and the possibility to intercalate ions between the layers [10,11].

Among these characteristics, bulk V_2_O_5_ has a low diffusion coefficient and poor electron transport [12]. Therefore, the formation of V_2_O_5_ nanostructures is crucial to improving the charge transport in this material. Several morphologies of V_2_O_5_ have been synthesized in order to obtain a nanostructured form, such as thin films, nanosheets, nanobelts, nanowires, nanotubes, or nanoparticles [13,14,15,16,17,18]. Furthermore, one-dimensional nanostructures demonstrate the best relationship between charge mobility and specific area. Moreover, these long-aspect-ratio nanostructures present a short radial diffusion channel for the interaction charges with the surrounded media and a suitable transport pathway for the electrons along the growth direction [19].

An interesting way to obtain one-dimensional V_2_O_5_-based materials has been through the implementation of templates, such as nanoporous alumina membranes [20], polycarbonate membranes [21], and carbon nanotubes (CNTs). CNTs have been successfully used, mainly in suspended form, as a template to form one-dimensional V_2_O_5_ nanostructures, either by removing the CNTs by thermal oxidation [22,23] or by keeping them, forming CNT/V_2_O_5_ composites [24,25]. For instance, the implementation of crystalline V_2_O_5_ attached to electrodes with good mechanical adhesion and high conductivity is of great importance to optimize the reversible ion exchange in electrochemical or photoelectrochemical systems [26].

Vertically aligned carbon nanotubes (VACNTs) form a remarkable material. They are helpful to preserve the properties mentioned above [27] and are also useful as a template with a high surface area that an oxidation process can remove. Thermogravimetric analysis (TGA) of the oxidative decomposition of VACNTs in an oxygen environment reveals that the temperature at which carbon gasification initiates is around 400–500 °C, depending on factors such as crystallinity and morphology [28,29,30,31]. On the other hand, V_2_O_5_ is a well-known catalyst for several compounds’ oxidation, distinguishing itself by exhibiting lower operation temperatures [32,33]. Therefore, to fabricate a CNT/V_2_O_5_ hybrid nanostructure, a trade-off between synthesis parameters arises due to an interaction between V_2_O_5_ and VACNTs in the oxidation process.

This work presents a reproducible procedure that allows the fabrication of several sponge-like V_2_O_5_ nanostructures using VACNTs as a template. By tuning the synthesis conditions, the template can be preserved or removed in its entirety. The specific vibrational modes, detected by Raman spectroscopy, ensure that the samples oxidized at 400 °C result in a hybrid CNT/V_2_O_5_ nanomaterial that can superpose the properties of V_2_O_5_ and the CNT matrix [34]. On the other hand, through oxidation at 500 °C and by varying the vanadium thickness, it is possible to control the formation of V_2_O_5_ nanofibers or CNT/V_2_O_5_ composites. The development of multi-purpose supported CNT/V_2_O_5_ nanostructures can help in the design of new battery electrodes [35], solid-state supercapacitors [36], and optoelectronic sensors [37].

## 2. Materials and Methods

### 2.1. Synthesis of VACNT

The VACNTs were grown by chemical vapor deposition (CVD) using as a catalyst a Fe/Al bilayer deposited onto n-type silicon substrates (100) by electron beam evaporation (e-beam) [38,39,40]. Slugs of Fe with 99.95% purity (Kurt Lesker, Jefferson Hills, PA, USA) and Al with 99.999% purity (Alfa Aesar, Haverhill, MA, USA) were located in FABMATE^®^ crucibles (Kurt Lesker) as source materials. First, a buffer layer of Al with 20 nm of thickness was deposited on the Si substrate using a deposition rate of 0.4 Å/s, and then 5 nm of Fe was deposited at 0.2 Å/s. Both layers were evaporated in the same vacuum cycle, with the chamber pressure kept below 2 × 10^−6^ Torr during evaporation, and a base pressure in the range of 10^−7^ Torr.

The VACNTs were grown to follow a four-step routine in a CVD system composed of a 1 m length and 0.06 m diameter quartz tubular horizontal furnace. First, the Fe/Al/Si substrates were located in the center of the quartz tube, that was heated up to 700 °C under Ar gas flow of 200 sccm, with a rate of 23 °C per minute. When the temperature set point was reached, the Ar flow was decreased to 150 sccm, and an H_2_ flow of 80 sccm was added by 20 min to promote the nucleation of the Fe layer in a reductive atmosphere. Then, a C_2_H_2_ flow of 50 sccm was added for 30 min to grow the VACNTs. Finally, the furnace was cooled down to room temperature using an Ar flow of 200 sccm.

### 2.2. Synthesis of V_2_O_5_/VACNTs

VACNTs were covered with different thicknesses of vanadium films (from 50 nm to 200 nm) deposited by e-beam evaporation using vanadium 99.7% purity (Varian, Palo Alto, CA, USA) with a deposition rate of 1 Å/s at a pressure below 2 × 10^−6^ Torr (base pressure in the range of 10^−7^ Torr). Then, to obtain V_2_O_5_, the samples were thermally oxidized under an O_2_ atmosphere in a horizontal quartz tube furnace at two temperatures: 400 °C and 500 °C. The oxidation time was 60 min, using a constant O_2_ flow of 100 sccm. The choice of these values arises from the study of the decomposition of the VACNT while vanadium oxide is formed [11,22]. A scheme of the entire synthesis process is shown in Figure 1.

### 2.3. Characterization

The materials were characterized by scanning electron microscopy (SEM) and energy-dispersive X-ray (EDX) spectroscopy using a scanning electron microscope from the company FEI, model Quanta 250 FEG. Structural analysis was performed with Raman spectroscopy using a Witec Alpha 300 RA system equipped with a low power 633 nm wavelength excitation laser to prevent photoluminescence effects and thermal damage in the samples. Finally, the samples were analyzed by X-ray photoelectron spectroscopy (XPS) using a FlexPS system from the company SPECS, equipped with a hemispherical analyzer model PHOBIOS 150 and a detector 1D-DLD, with a monochromatic X-ray source model FOCUS 500 providing Al ka radiation with a characteristic energy of 1486.71 eV.

## 3. Results

Figure 2 shows SEM micrographs of the top view and cross-section of a set of samples with 50 nm and 200 nm of vanadium thickness deposited on VACNTs and subsequently oxidized at 400 °C or 500 °C. Figure 2a shows a top-view micrograph of the sample with 50 nm of V and oxidized at 400 °C. Their morphology is similar to that of the pristine VACNT sample (see Appendix A); however, in this case, the nanostructures have a thicker diameter with some irregularities in their walls and in some regions they look to be as if welded between themselves. The sample fabricated with 200 nm of V and oxidized at 400 °C (Figure 2b) exhibits bigger diameter nanostructures with a faceted grain shape. This lamellar growth is consistent with that observed in V_2_O_5_ structures [11,41]. In the case of samples oxidized at 500 °C, Figure 2c,d, the structures do not have a tubular shape and exhibit well-defined faceted grains, with the sample fabricated with 200 nm of V presenting larger grains. Top-view micrographs of samples fabricated with 100 nm and 150 nm of V and oxidized at both temperatures are shown in Appendix A. These samples have a morphology consistent with the previous discussion.

From the side-view micrographs, it is possible to observe that in samples oxidized at 400 °C (Figure 2e,f), the V_2_O_5_ nanostructures are mostly in the top portion of the CNTs. However, in the case of the samples oxidized at 500 °C (Figure 2g,h), the presence of the CNTs is strongly reduced. Particularly in the sample with 50 nm of V oxidized at 500 °C, it is not possible to observe the presence of CNTs. This observation is also confirmed by the EDX results.

The elemental composition of the samples was analyzed using energy-dispersive X-ray spectroscopy (EDX) with an electron acceleration voltage of 15 kV. Figure 3a,b show the spectra of all samples oxidized at 400 °C and 500 °C, respectively. As expected, due to the high depth penetration of the electrons at 15 kV, the main signal detected at 1.74 keV is the silicon Kα line that belongs to the substrate. Additionally, the aluminum Kα contribution is observed at 1.49 keV, which corresponds to the metallic bilayer used as a catalyst in the growth of the VACNTs. The vanadium contribution is observed at peaks situated at 4.95, 5.43, and 0.51 eV, which correspond to the Kα, Kβ, and Lα lines, respectively, while oxygen and carbon Kα lines are located at 0.586 and 0.282 keV, respectively. In the case of samples oxidized at 500 °C and coated with 50 and 100 nm of vanadium, the Kα line of the carbon contribution is not observed. To further investigate the atomic concentration of V and C in the samples prepared under different conditions, the plots in Figure 3c,d were elaborated for each element, respectively. Figure 3c shows that for both oxidation temperatures, the weight percentages of V in the sample vary linearly with respect to the deposited V thickness. However, as shown in Figure 3d, the weight percentage of C exhibits a different behavior related to the oxidation temperature. In samples oxidized at 400 °C, the weight percentage of C drops from 74% to 52%, and in samples oxidized at 500 °C, this percentage is close to zero in the samples coated with 50 and 100 nm of vanadium. Meanwhile, in samples with thicker vanadium coatings (150 and 200 nm), the weight percentage of C is close to 17%. This remnant amount of carbon observed in the samples with thicker deposits of V (150 and 200 nm) could be attributable to a loss in the permeability of the oxygen through the thicker V_2_O_5_ films, reducing the oxidation rate of the VACNTs under the oxidation conditions mentioned above.

Figure 4 shows the Raman spectra of V-coated VACNT samples oxidized at 400 °C (Figure 4a) and 500 °C (Figure 4b). First, analyzing the range from 100 to 1000 cm^−1^, it is possible to notice that all spectra exhibit 10 peaks, which can be associated with the 14 vibrational modes of the V_2_O_5_ polymorph (7A_g_ + 3B_3g_ + 3B_1g_ + 1B_2g_). This indicates that the crystallization of vanadium oxide is in the orthorhombic V_2_O_5_ phase. The peaks located at 102, 144, and 198 cm^−1^ represent the external bending vibration modes A_g_, B_1g_/B_3g_, and A_g_/B_2g_, respectively. These vibrations are related to the motion of the V_2_O_5_ layers between each other, and the higher intensity of the 144 cm^−1^ resonance indicates that crystallization occurs preferentially along to c-axis of this phase [42]. The peaks at 302, 402, 480, and 522 cm^−1^ represent the internal bending vibration modes A_g_, and peaks at 283 and 697 cm^−1^ represent the internal stretching vibrations B_1g_/B_3g_ [43]. On the other hand, the peak located at 991 cm^−1^ represents the external V^5+^=O (vanadyl) A_g_ vibrational mode. In all samples, this peak is situated in a lower wavenumber than in bulk material (~995 cm^−1^), which has been previously attributed to slight deviations in the stoichiometry of the phase due to oxygen vacancies [44,45,46].

Analyzing the range between 1000 and 1900 cm^−1^ of the spectra in Figure 4a (samples oxidized at 400 °C) it is possible to observe that the sample coated with 50 nm V exhibits the characteristic vibrational modes related to the CNTs (broad peaks located around 1320 and 1590 cm^−1^) with a similar intensity to the V_2_O_5_ peaks. However, the intensity of these CNT peaks decreases dramatically when the V coating thickness increases. In the case of samples oxidized at 500 °C (Figure 4b), these modes are not present in the spectra.

These results, together with the previous analysis of the SEM and EDX characterizations, indicate that vanadium strongly affects the oxidation process of the VACNT matrix. To explore this effect, Raman spectra were measured in the ranges where CNTs exhibit their characteristic resonances, on the sample coated with 50 nm V and oxidized at 400 °C (labeled as “VACNT_50V_400”), together with the other three samples: as-grown VACNTs (labeled as “VACNT”), VACNTs without vanadium oxidized in an O_2_ atmosphere at 400 °C (labeled as “VACNT_400”), and another VACNT sample oxidized at 500 °C (labeled as “VACNT_500”). These spectra, shown in Figure 5a, are characteristics of CNTs with low crystallinity, with the exception of the spectrum of the sample with the V coating, indicating the existence of CNTs with a higher graphitization level [47,48,49]. The following quantitative analysis, based on the fit of these spectra, supports this observation.

In the 1000–1750 cm^−1^ region, the fit was performed using the five-peak model for carbonaceous materials proposed by Sadezky and coworkers [50], which consider 4 Lorentzian + 1 Gaussian component. The Fityk software with the Levenberg–Marquardt model was used, and an accuracy for the fits of R^2^ > 0.98 was required [51]. The G band, generated by bond-stretching motion of the C–C pair (E_2g_ vibrational mode in graphite), is observed at 1590 cm^−1^ for the VACNT and VACNT_400 samples, and is located around 1584 cm^−1^ for the VACNT_500 and VACNT_50V_400 samples (see Figure 5b). According to the “three-stage model” (TSM) proposed by Ferrari and Robertson [52,53], the shift of the G peak position to lower wavenumbers is an indication of the reduction in the degree of the structural disorder in this kind of carbonaceous material (located in the so-called second stage). On the other hand, the D band, associated with an active A_1g_ breathing mode in carbon structures [54], is observed at 1320 cm^−1^ for the VACNT and VACNT_400 samples, around 1328 cm^−1^ for VACNT_500, and around 1330 cm^−1^ for VACNT_50V_400. The position of this peak has been reported to be influenced by the excitation wavelength [55] and also by the diameter of the CNTs [56]. In this case, the shift to higher wavenumber values could be explained by a reduction in the diameter of CNTs due to the carbothermic decomposition in the oxidation process. The full width at a half maximum, FWHM, of the G and D peaks (Figure 5c) can also be used as an indicator of the structural disorder degree [47]. In this case, the smaller value for the FWHM of both peaks is exhibited by the spectrum of VACNT_50V_400, which is also an indication of a reduction in the amorphization in the VACNTs. The intensity ratio between the D and G peaks (I_D_/I_G_) is shown in Figure 5d. For the VACNT and VACNT_400 samples, the value of this ratio is around 1.1; it increases up to 1.3 for the VACNT_500 sample, and the VACNT_50V_400 sample exhibits a higher value around 1.6. According to the TSM (second stage) this is another indicator that samples with a high value of I_D_/I_G_ present the lowest degree of amorphization, or equivalently, the highest degree of graphitization. Also, the best definition of the D’ peak in the VACNT_50V_400, observed at ~1610 cm^−1^ and associated with a second-order vibration of the D band [57], confirms the highest degree of graphitization of this sample.

The peaks labeled as 7A_1_ and 5A_1_, located at ~1200 and ~1500 cm^−1^, respectively, are related to poly-aromatic vibrations, commonly due to the presence of amorphous carbon, polyenes, or poly-aromatic hydrocarbons (PAHs) [47,50,58]. Therefore, the intensity ratio between these signals and the D band (I_5A1_/I_D_ and I_7A1_/I_D_) are also indicators of disorder in the carbon honeycomb structure of the CNTs. In this case, see Figure 4e, these indicators show that vanadium’s presence improves the graphitization of the VACNTs more than the thermal oxidation process at 400 °C or 500 °C. Finally, in the range of the second-order vibration bands of CNTs, it is possible to distinguish at ~2645 cm^−1^ the D band and at ~2900 cm^−1^ the D + G band [57]. The 2D band is better defined in the sample VACNT_50V_400 and the intensity ratio between this band and the D band is also highest in this sample (Figure 5e), reflecting their best degree of graphitization. To complement the previous analysis, XPS was realized to obtain information about the oxidation state of C atoms in the samples (survey spectra are shown in Appendix A). Figure 6 shows the high-resolution spectra of the C1s signal of samples VACNT_50V_400 and VACNT_50V_500, Figure 6c and Figure 6e, respectively, together with VACNT_400 and VACNT_500 (Figure 6b,d) and the VACNT. The fitting of the C1s signal of each sample was made considering six contributions related to C=C sp^2^, C-C sp^3^, C-O, C=O, O-C=O, and π-π* bonds. The parameters of the fit are shown in Table 1. As is expected, the spectra of VACNT and VACNT_400 are very similar. However, in the sample VACNT_50V_400, an increment in the contributions related to the formation of oxygen bonds is observed. The intensity of these signals also increases in sample VACNT_500 and is particularly strong in sample VACNT_50V_500, which exhibits the largest contributions of oxygen–carbon bonds.

These results confirm the catalytic behavior of the vanadium during the oxidation procedure. Under these conditions, oxygen atoms can penetrate the walls of nanostructures and initiate the decomposition of carbon. Due to the fact that the decomposition of amorphous carbon is energetically favored, the presence of defects in the CNT walls is decreased, reducing the CNT wall thickness and at the same time improving their graphitization [58]. This catalytic effect is responsible for the removal of the carbon template in the samples with 50 nm and 100 nm of vanadium oxidized at 500 °C [59].

Finally, to investigate the chemical composition of the vanadium oxide species formed in the oxidation treatment, the high resolution of the V_2p_ region was measured in the samples VACNT_50V_400 and VACNT_50V_500, shown in Figure 7a,b, respectively. In both spectra the characteristic peaks are observed related to the V2p doublet, V2p_3/2_, and V2p_1/2_, at binding energies of 517.0 eV and 524.5 eV, respectively, which corresponds to the V^5+^ oxidation state in V_2_O_5_. However, both peaks exhibit an asymmetric shape with a weak shoulder at low binding energy, which suggests the presence of a minor percentage of V atoms with a lower oxidation state. These spectra were adjusted by two pairs of doublets using Gaussian–Lorentz curves, the aforementioned V^5+^ doublet (red curves) and the peaks (blue curves) located at 515.8 eV (2p_3/2_) and 522 eV (2p_1/2_). This analysis suggests the formation of a small quantity of V^4+^ [60], which is consistent with the Raman characterization, due to the fact that the Raman spectra of both samples exhibit a blueshift of the vanadyl peak that could be related to the formation of oxygen vacancies in the V_2_O_5_ structure. Finally, a rough analysis of the concentration ratios of the V species was performed; the sample oxidized at 400 °C shows a V^5+^:V^4+^ concentration of 91:9, whereas in the sample oxidized at 500 °C, the V^5+^:V^4+^ concentration is 95:5. The formation of V^4+^ species is strongly associated to the carbothermic reduction process, which induces the formation of oxygen vacancies in the V_2_O_5_ structure and can also be considered a factor in improving the electrical conductivity of the samples [25,61].

## 4. Conclusions

This manuscript reports a way to fabricate sponge-like V_2_O_5_ nanostructures based on VACNTs used as templates. According to the thermal oxidation temperature and thickness of the V coating, it is possible to obtain a supported sponge-like nanostructure composed of VACNT/V_2_O_5_ with improved graphitization or just V_2_O_5_ nanofibers. It was identified that there is a limit to which the amount of deposited vanadium acts as a stopper, inhibiting the oxidation process of the VACNTs as well as the correct diffusion of oxygen inside the structure. The XPS spectra confirm that V_2_O_5_ improves the carbothermic reduction by the formation of C-O species more than the pristine VACNTs. This effect can be attributable to a reduction in the wall thickness due to carbon decomposition. The interaction of V_2_O_5_ with VACNTs during the oxidation also leads to the formation of V^4+^ and related oxygen vacancies. The presence of these defects holds promising applications in enhancing the performance of the nanostructure, especially when utilized as an ion-intercalation material [62,63,64]. Finally, Raman analysis reveals that the early stages of the VACNT oxidative decomposition involve amorphous carbon etching, resulting in an improvement in the graphitization of the CNTs contained in the templates. These findings, in conjunction with the high surface area of this sponge-like hybrid nanostructure, suggest a potential use in electrochemical applications, particularly as a cathodic intercalation material in lithium-ion batteries.

## Figures and Tables

**Figure 1 nanomaterials-14-00211-f001:**
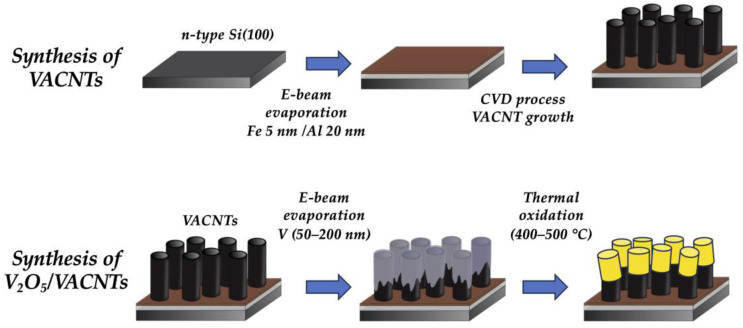
Scheme of the synthesis process of VACNTs and V_2_O_5_/VACNTs.

**Figure 2 nanomaterials-14-00211-f002:**
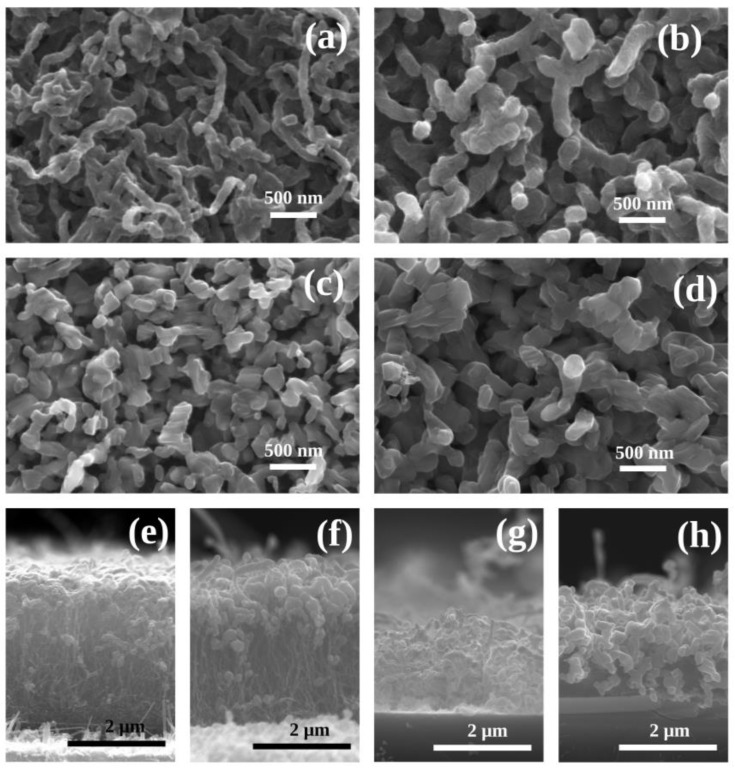
SEM micrographs. Top view of samples with (**a**) 50 nm and (**b**) 200 nm of V, oxidized at 400 °C. Top views of samples with (**c**) 50 nm and (**d**) 200 nm of V, oxidized at 500 °C. Lateral view of samples with (**e**) 50 nm and (**f**) 200 nm of V, oxidized at 400 °C. Lateral view of samples with (**g**) 50 nm and (**h**) 200 nm of V, oxidized at 500 °C. The scales of 500 nm and 2 µm correspond to (**a**–**d**) and (**e**–**h**), respectively.

**Figure 3 nanomaterials-14-00211-f003:**
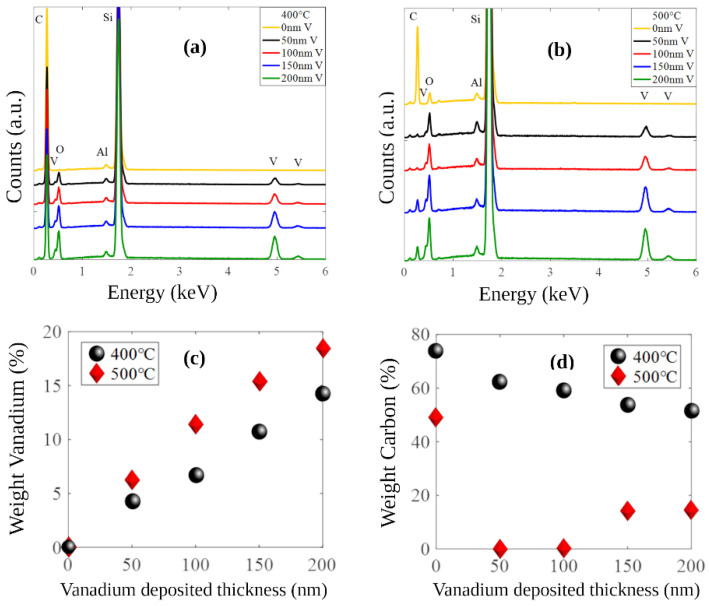
EDX spectra of VACNT samples, loaded with 0, 50, 100, 150, and 200 nm V, oxidized at (**a**) 400 °C and (**b**) 500 °C. Weight percentages of (**c**) V and (**d**) C as a function of vanadium thickness in samples oxidized at 400 °C and 500 °C.

**Figure 4 nanomaterials-14-00211-f004:**
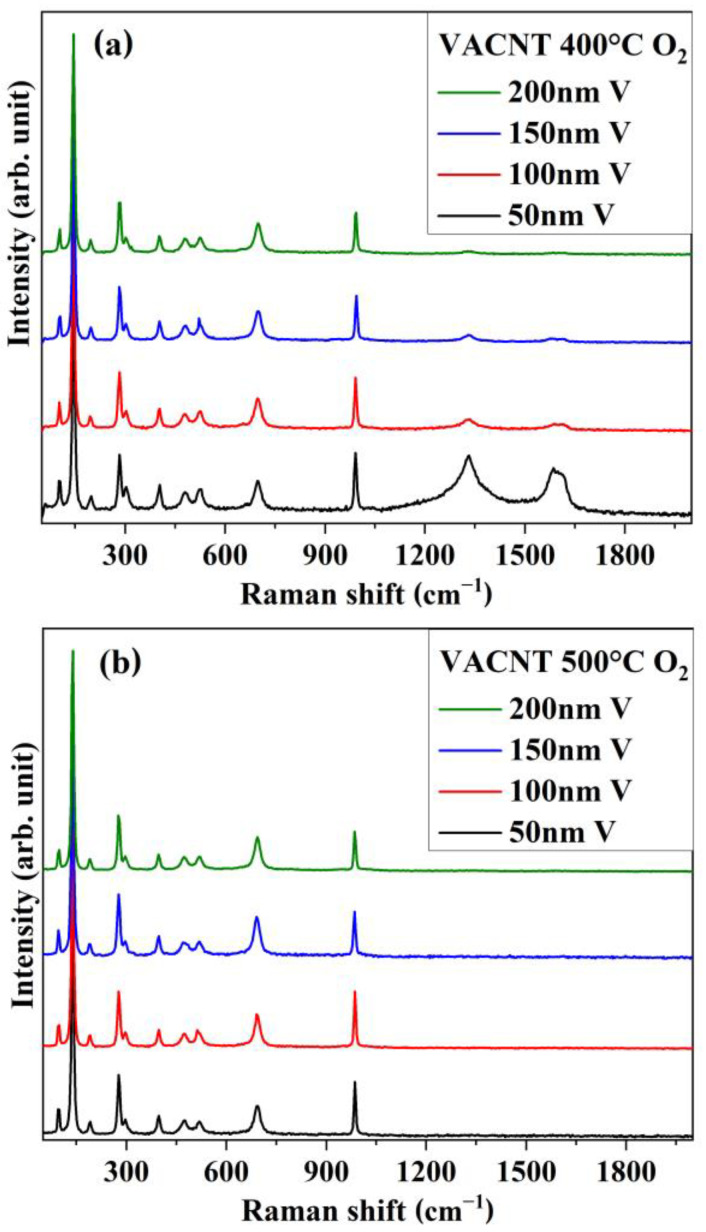
Raman spectra of V-coated VACNT samples oxidized in an O_2_ atmosphere at (**a**) 400 °C and (**b**) 500 °C.

**Figure 5 nanomaterials-14-00211-f005:**
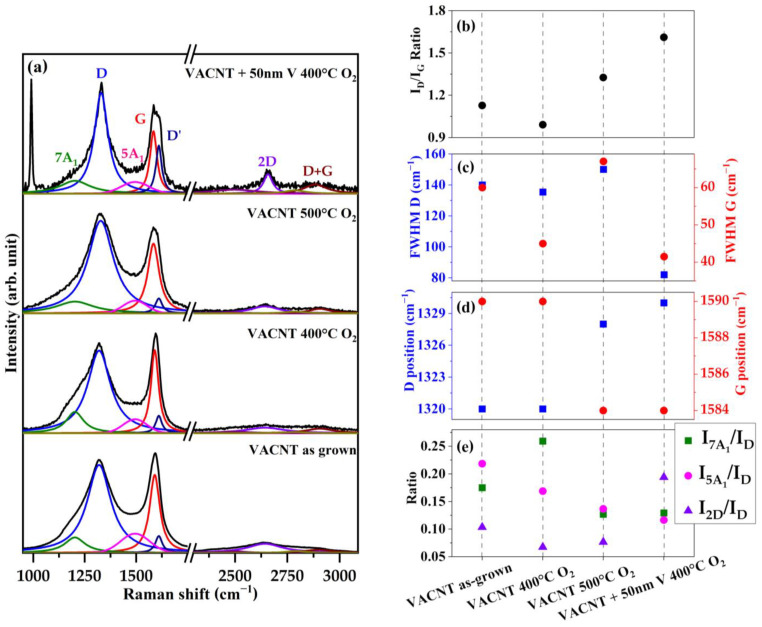
(**a**) Carbon region Raman spectra of the VACNT/V_2_O_5_ sample compared with VACNTs as-grown and VACNTs with different thermal annealing processes. (**b**) The ratio of the intensities of the D and G peaks, (**c**) FWHM of the D and G peaks, (**d**) Raman shift position of the D and G peaks, and (**e**) intensity ratio between the 7A_1_, 5A_1_, and 2D peaks over the D peak.

**Figure 6 nanomaterials-14-00211-f006:**
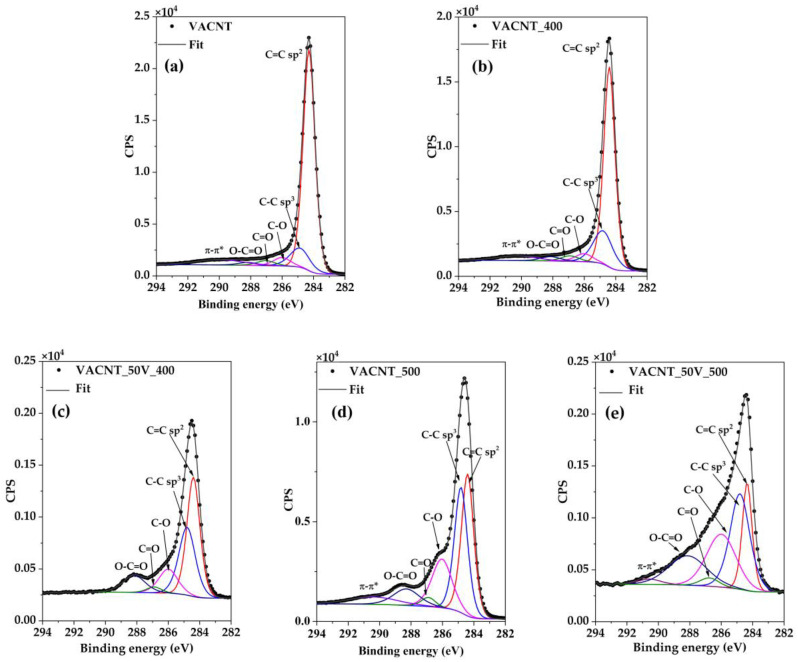
High-resolution XPS spectra of the C1s signal of: (**a**) As grown VACNTs, (**b**) VACNTs oxidized at 400 °C, (**c**) VACNTs with 50 nm V oxidized at 400 °C, (**d**) VACNTs oxidized at 500 °C, (**e**) VACNTs with 50 nm V oxidized at 500 °C.

**Figure 7 nanomaterials-14-00211-f007:**
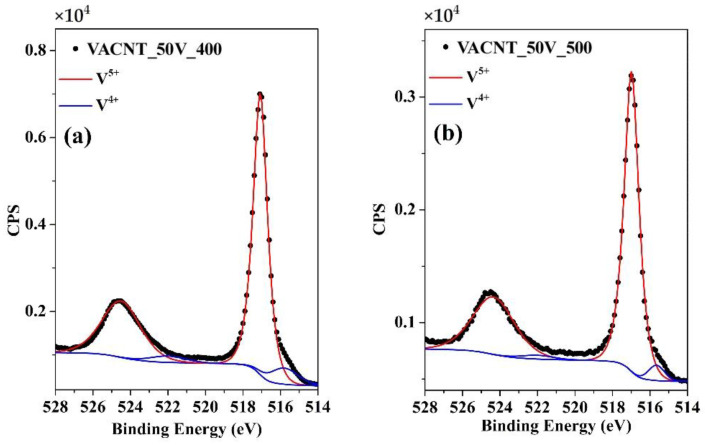
(**a**) and (**b**) correspond to the high-resolution XPS of the V2p signal for samples VACNT_50V_400 and VACNT_50V_400, respectively. The red doublet represents the V^5+^ contribution, while the blue doublet represents the V^4+^ contribution.

**Table 1 nanomaterials-14-00211-t001:** Fitting parameters of the C1s signal, showing the position (Pos) and the concentration (Con) of each contribution.

	VACNT		VACNT_400		VACNT_50V_400		VACNT_500		VACNT_50V_500	
	Pos (eV)	Con %	Pos (eV)	Conc %	Pos (eV)	Con %	Pos (eV)	Con %	Pos (eV)	Con (%)
C=C sp^2^	284.3	74.0	284.4	65.8	284.4	43.0	284.4	33.3	284.3	18.2
C-C sp^3^	284.9	10.8	284.8	19.4	284.8	31.2	284.8	31.1	284.8	32.3
C-O	286.0	4.0	286.0	4.1	286.0	12.7	286.0	18.5	286.0	26.0
C=O	287.0	2.5	286.9	2.9	286.9	2.5	286.9	2.4	286.8	2.6
O-C=O	288.2	1.6	288.1	2.5	288.2	10.6	288.3	8.1	288.1	18.8
π-π*	290.0	7.0	290.3	5.3	N/A	0	290.4	6.6	920.5	2.1

## Data Availability

S.H. is the depository of all the data generated by the study.

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
