# Peer review of "The Synthesis of Sponge-like V2O5/CNT Hybrid Nanostructures Using Vertically Aligned CNTs as Templates"

_nanomaterials, 2024, doi:10.3390/nano14020211_

Round 1

Reviewer 1 Report

Comments and Suggestions for Authors

The manuscript presents the fabrication of sponge-like V2O5/CNT nanostructure. The structural and composition analysis illustrates the impact of oxidation temperature and vanadium thickness, and  confirms vanadium as the catalyst to decompose CNT. Minor revision is suggested. There are a few suggestions that the author may consider:

1. For Figure 3b, C wt% is close to 0 for 50nm and 100nm vanadium, while C wt% is higher for 150nm and 200nm vanadium. Could the authors provide some explanation? For Figure 4b, why CNT peaks (at 1320 and 1590 cm-1) are not visible for 150nm and 200nm vanadium?

2. V2O5 nanofibers has been extensively studied. The manuscript will be improved if the authors can elaborate the novelty.

Minor problems:

Grammar should be checked. For example:

Page 5: Raman spectra were measured in the ranges were where CNTs exhibit their characteristic resonances

Page 6: the shift to higher wavenumber values could be explain explained by a reduction of the diameter 

Page 7: this is other indicator that samples with a high value of ID/IG presents the less degree of amorphization 

Comments on the Quality of English Language

Minor grammar editing is encouraged

Reviewer 2 Report

Comments and Suggestions for Authors

Comments on the Quality of English Language

The english language looks average, with many typos
